# Genome-Wide Selective Analysis of Boer Goat to Investigate the Dynamic Heredity Evolution under Different Stages

**DOI:** 10.3390/ani12111356

**Published:** 2022-05-26

**Authors:** Ying Yuan, Weiyi Zhang, Chengli Liu, Yongmeng He, Haoyuan Zhang, Lu Xu, Baigao Yang, Yongju Zhao, Yuehui Ma, Mingxing Chu, Zhongquan Zhao, Yongfu Huang, Yanguo Han, Yan Zeng, Hangxing Ren, Gaofu Wang, Guangxin E

**Affiliations:** 1College of Animal Science and Technology, Southwest University, Chongqing 400715, China; 15703001951@163.com (Y.Y.); lwing0829@163.com (W.Z.); lcl222333@outlook.com (C.L.); yongmenghe123@163.com (Y.H.); swuzhanghy@163.com (H.Z.); lujiusym@163.com (L.X.); yangbaigao915@163.com (B.Y.); h67738337@swu.edu.cn (Y.H.); hyglyp1987@163.com (Y.H.); zengyan@swu.edu.cn (Y.Z.); 2Chongqing Key Laboratory of Forage & Herbivore, Chongqing 400715, China; zyongju@163.com (Y.Z.); zhongquanzhao@126.com (Z.Z.); 3Institute of Animal Husbandry and Veterinary Medicine, Chinese Academy of Agricultural Sciences, Beijing 100097, China; mayuehui@caas.cn (Y.M.); chumingxing@caas.cn (M.C.); 4Chongqing Academy of Animal Sciences, Chongqing 402460, China; rhxe@163.com (H.R.); 20031216@163.com (G.W.)

**Keywords:** Boer goat, genome-wide selective analysis, artificial selection, candidate genes

## Abstract

**Simple Summary:**

Boer goat is a world-famous meat-type goat breed that underwent long-term artificial selection from African indigenous animals. The current study displayed the genome-wide selection signature analyses of South African indigenous goat (AF), African Boer (BH), and Australian Boer (AS), to investigate the hereditary basis of artificial selection in different stages. Moreover, the θπ, F_ST_, and XP-CLR methods were used to search for the candidate signatures of positive selection in Boer goats. Ten genes (e.g., *BMPR1B*, *DNER*, *ITGAL*, and *KIT*) under selection in both groups were identified and are potentially responsible for reproduction, metabolism, growth, and development. This study provided a comprehensive overview of genomic variations in Boer goat, which may provide a basis for further resource protection and improvement of this breed.

**Abstract:**

Boer goats, as kemp in meat-type goats, are selected and bred from African indigenous goats under a long period of artificial selection. Their advantages in multiple economic traits, particularly their plump growth, have attracted worldwide attention. The current study displayed the genome-wide selection signature analyses of South African indigenous goat (AF), African Boer (BH), and Australian Boer (AS) to investigate the hereditary basis of artificial selection in different stages. Four methods (principal component analysis, nucleotide diversity, linkage disequilibrium decay, and neighbor-joining tree) implied the genomic diversity changes with different artificial selection intensities in Boer goats. In addition, the θπ, F_ST_, and XP-CLR methods were used to search for the candidate signatures of positive selection in Boer goats. Consequently, 339 (BH vs. AF) and 295 (AS vs. BH) candidate genes were obtained from SNP data. Especially, 10 genes (e.g., *BMPR1B*, *DNER*, *ITGAL*, and *KIT*) under selection in both groups were identified. Functional annotation analysis revealed that these genes are potentially responsible for reproduction, metabolism, growth, and development. This study used genome-wide sequencing data to identify inheritance by artificial selection. The results of the current study are valuable for future molecular-assisted breeding and genetic improvement of goats.

## 1. Introduction

Following the advances of high-throughput sequencing technology, the scientific community has widely explored relative heredity basis with adaptability and economic phenotype of animals using genome wide selective analysis strategy (GWSA) [1,2,3]. In particular, a series of candidate genes related to domestication and artificial selection of domestic animal was determined. For example, the mutation genotype pattern located in the promoter region of *PDGFD* may be the cause of fat deposition in the tail of sheep [4]; the AHR gene related to female reproduction is under strong positive selection during the domestication of pigs [5]. An 8-kb sequence spanning the *AMY2B* locus showed signals of selection of key roles in starch digestion and fat metabolism [6].

Furthermore, many studies displayed the GWSA to investigate the key genes of environmental adaptability [7] and multiple economic traits [8,9] in goats, as well as their population migration [10], phylogeny, and domestic history [11]. For instance, a study found that *MUC6* gene mutation in goats can improve the antiparasitic ability of their gastrointestinal tract to ensure that they have more adaptability to the human environment [12].

According to relevant reports, the indigenous goat accounted for more than 63% of total stock in South Africa, and it displayed a remarkable contribution to local livestock husbandry. Moreover, high-performance breeds under artificial selection appeared with the transformation of small-scale production systems to large-scale commercial agriculture [13]. Boer goat is a world-famous meat-type goat breed that underwent long-term artificial selection from African indigenous animals, and Australian Boer goats are introduced from Africa and further artificially selected [14]. To date, many countries have introduced Boer goats to improve local breeds and achieved excellent benefit. However, there is very little research literature on genomic changes in Boer goats. Therefore, the genetic basis of artificial selection in the improvement of economic traits can be elucidated by investigating the genomic genetic divergence between Boer goats and African indigenous goats with different stages.

In this study, we displayed the GWSA among African indigenous, South African Boer, and Australian Boer goats to identify the inheritance shake by artificial selection. Results were valuable for future molecular-assisted breeding and genetic improvement of goats.

## 2. Materials and Methods

### 2.1. Animals, Genome Sequencing, and Data Acquisition

The experiment was carried out in strict accordance with the guidelines of the International Cooperation Committee of Animal Welfare on the care and use of experimental animals.

Genomic DNA of 10 African Boer goats (BH) were supplied by Sokoine University of Agriculture (SUA) of Tanzania. Sequencing libraries of all samples were constructed by using NEBNext^®^ ΜLtra DNA library preparation kit (Illumina^®^, 15,026,486 Rev. C, San Diego, CA, US) and whole genome sequencing was performed by Illumina NovaSeq 6000 × Ten platform (BGI, Shenzhen, China), the sequencing depth, for BH animals, was expected around 10×. The genome datasets of total 30 Australian Boer goats (AS) from our previous study [15] and 30 African indigenous goats (AF) were obtained from NCBI SRA database (PRJNA671542), respectively (Appendix A).

### 2.2. Read Filtering, Alignment, and Variant Calling

The quality controlled and filtered raw sequencing reads (RSR) were obtained by fastp (v0.20.1)”—w 15—cut—window—size 4—cut—mean—quality 15—5 3—3 3—length—required 40”, which is a method used to filter out low quality reads (HQRs). The HQRs were mapped to the *Capra hircus* reference genome of ARS1 (GCF_001704415.1) and the MEM algorithm of Burrows–Wheeler Aligner (v.0.7) was selected with the minimum mapping quality set to 20 and unmapped reads were filtered out using SAMtools (v.1.3). Following this, read groups were added and PCR duplicates were marked using Picard (http://broadinstitute.github.io/picard/ (accessed on 1 June 2021)). Single nucleotide polymorphism (SNP) variants were called by GATK (v.3.7) with HaplotypeCaller, Genotype GVCFs, and Select Variants module. After SNP calling, we used the module “Variant Filtration” of GATK to obtain high-quality SNPs with the parameters. Variants were filtered based on “QD < 2.0 || FS > 200.0 || SOR > 10.0 || MQRankSum < −12.5 || ReadPosRankSum < −8.0”. After filtering, the SNPs were filtered out again by “—max-missing 1—max-alleles 2—min-alleles 2—maf 0.05”.

### 2.3. Diversity, Phylogenetic, and Population Genetic Analyses

Distance matrix was calculated by VCF2D is (https://github.com/BGI-shenzhen/VCF2Dis (accessed on 1 June 2021)) and Neighbor-joining phylogenetic network was constructed by Phylip (https://github.com/topics/phylip (accessed on 1 June 2021)), and visualization performed by iTOL online tool (https://itol.embl.de/ (accessed on 1 June 2021)). Principal component analysis (PCA) was estimated and graphics visualized by GCTA (https://github.com/cooljeanius/gcta (accessed on 13 July 2021)) and R program (ggplot2 package), respectively. Linkage disequilibrium (LD) was performed using Pop LDdecay software (https://github.com/BGI-shenzhen/Pop LDdecay (accessed on 13 July 2021)).

### 2.4. Genome-Wide Selective Sweep Analysis and Gene Annotation

GWSA of SNPs were performed with two groups, as follows: (1) 10 African Boer goats (case) versus 30 African indigenous individuals (control); and (2) 30 Australian Boer goat (case) versus 10 African Boer goats (control). For the SNPs dataset, the pairwise fixation index (F_ST_) and π ratio (π_case_/π_control_) were calculated with 40 kb sliding windows and 20 kb step size using vcftools [16]. The cross-population composite likelihood ratio (XP-CLR) is a method for the determination of the selection signal, can avoid ascertainment biases and successfully detect older signals and the selections on standing variation, which means the allele frequency changed very rapidly due to random drift between two populations. Non-overlapping sliding windows of 50 kb were used, and the maximum number of SNPs within each window was 600 [17]. The top 5% was chosen as the significance threshold for XP-CLR. The variants were annotated with ANNOVAR (ANNOate VARiation), which can utilize annotation databases from annotation data set conforming to Generic Feature Format version 3 (GFF3). Candidate genes were annotated by the intersection of the common parameters with top 5% (F_ST,_ πratio, XP-CLR). We can estimate the fraction of intronic, exonic, missense, stop-gained, etc. variants after annotated by Variant Effect Predictor (VEP). Additionally, candidate genes were subjected to gene ontology (GO) and Kyoto Encyclopedia of Genes and Genomes (KEGG) with KOBAS 3.0 (http://kobas.cbi.pku.edu.cn/ (accessed on 28 July 2021)) rely on human.

## 3. Results

Obtained from 70 animals were 18,321,865 SNPs. The frequency of most SNPs is located in the intergenic (66.79%) and intron (26.66%) regions. The Ti/TvRatio is 2.38. On the contrary, the least frequency of SNPs is distributed in the 3′ UTR (0.32%) and 5′ UTR (0.07%; Appendix A).

The phylogenetic network revealed that all populations were divided into single linkage (Figure 1A), which corresponded to that of the PCA pattern (Figure 1B). Analysis result of LD (*r*^2^) showed the highest mean LD and the slowest decay in the AS population. In contrast, the lowest mean LD and the fastest decay in the AF population (Figure 1C) are shown. Coincidentally, the highest genome-wide nucleic acid diversity (ND) is AF in comparison with BH and AS (Figure 1D).

The GWSA result of group 1 (BH vs. AF) revealed that 339 candidate genes (Appendix A) were extracted from intersected windows by the top 5% of common parameters of F_ST_, θπ rate, and XP-CLR (Figure 2A). The GO analysis results (Appendix A) showed that a high growth trait-related GO terms were identified, such as tissue development (GO:0009888), muscle fiber development (GO:0048747), and fibroblast growth factor binding (GO:0017134). In addition, several reproduction-related terms were also identified, such as positive regulation of intracellular estrogen receptor signaling pathway (GO:0033148), gonad development (GO:0008406), and sperm-egg recognition (GO:0035036). In addition, the KEGG analysis results (Appendix A) showed that a significant number of genes are enriched in the top 10 (corrected *p* value < 0.01) pathways (i.e., transcriptional misregulation in cancer, pathways in cancer, glutamatergic synapse, oocyte meiosis, human T-cell leukemia virus 1 infection, Rap1 signaling pathway, Ras signaling pathway, and progesterone-mediated oocyte maturation).

Subsequently, the result of GWAS in group 2 (AS vs. BH) using the SNP dataset revealed that 295 candidate genes (Appendix A) were obtained by the top 5% of three parameters (Figure 2B) at the gene overlapping region. It was similar to the GO and KEGG (Appendix A) enrichment in the rich multiple categories of group 1, such as those with well-known pathways (i.e., PI3K-Akt signaling pathway, Rap1 signaling pathway, MAPK signaling pathway, and Ras signaling pathway). Numerous GO terms were related to growth development (e.g., striated muscle tissue development, cardiac muscle contraction, and positive regulation of smooth muscle cell proliferation), reproduction (e.g., intracellular estrogen receptor signaling pathway, spermatogenesis, and male gonad development), immunity (e.g., T cell costimulation, T cell migration, and B cell apoptotic process), and metabolism (e.g., insulin secretion, glycogen catabolic process, and phosphate-containing compound metabolic process).

Interestingly, outstanding visible divergence in digestion and metabolism-related KEGG pathways was found according to the results of the gene functional enrichment of selected genes at two different stages (Table 1). Notably, 10 genes (*BMPR1B*, *DNER*, *EPHA6*, *ERCC6L2*, *ITGAL*, *KIT*, *PROM1*, *RBFOX2*, *RFX3*, and *U6*) under selection in both groups were identified.

## 4. Discussion

In this study, the ND of both Boer goat populations was lower than African indigenous goats. The higher LD linkage within the AS population indicated that the frequency of genotype may be gradually changed at different stages as humans expected during the artificial selection process [18,19,20]. The phylogenetic and PCA pattern relationship of the three populations was inconsistent with their geographic location, which implied that artificial selection may contribute to the changes in animal genome diversity greatly and rapidly.

Boer goat was initially selected and bred from AF goats and has a superior economic phenotype and fecundity [14,21]. Interestingly, 10 genes were identified to undergo selection in both breeding stages of the Boer goat, which was widely confirmed to be involved in various biological functions (e.g., reproduction, growth, and metabolism).

Despite a series of metabolism-related (MRD) genes found in the two stages of BH and AS, the genes and their MRD pathways were different. For example, in terms of amino acid metabolism, six pathways related to amino acid metabolism were enriched and involved in the digestion and absorption of 17 essential and nonessential amino acids in the AF to BH stages. In the BH to AS stage, only three MRD pathways were enriched including eight kinds of amino acids. The AF feeding conditions are generally single and backward [22]. In addition, BH has undergone artificial selection and breeding in the commercial system with improved living conditions and medical security. Corresponding changes have been made in the environmental adaptation, metabolism, and other hereditary traits of Boer goats. Furthermore, the differences in the ecological environment of the habitat, breeding standards, and improved raising level in AS led to the rapid coevolution of the metabolism and environment of Boer goats. This was also confirmed by a large number of candidate genes related to the metabolism of fatty acids, minerals, and vitamins, as identified in AS vs. BH. Thus, the coevolution between the metabolic inheritance and the integrated living environment in animals has been universally confirmed [23,24,25].

Particularly, some candidate genes related to metabolism have been attracting attention. Ectonucleotide pyrophosphatase/phosphodiesterase3 (*ENPP3*), as a regulator of *N*-acetylglucosaminyltransferase GnT-IX (GnT-Vb), would have widespread and significant impacts on glycosyltransferase activities [26]. Short-chain enoyl-CoA hydratase *(ECHS1*) is involved in amino and fatty acid catabolism in mitochondria and its deficiency results in Leigh syndrome or exercise-induced dystonia [27]. Niemann–Pick C1-Like 1 (*NPC1L1*) was widely verified to be a cholesterol import protein and mediated intestinal cholesterol absorption [28]. It also participated in the intestinal absorption of fat-soluble vitamins [29]. In addition, genes that have been continuously selected in terms of immune resistance exist due to changes in the living environment. The integrin LFA-1 (also known as *ITGAL*), also known as *CD11a*, is upregulated in metastatic melanoma, highly expressed in most immune cell populations [30]. *ITGAL* is closely linked to the pathogenesis of diverse immune-related diseases [31]. Therefore, the capture of these genes implied that the metabolic and physiological functions of Boer goats are continuously affected by the selective breeding process.

A series of reproduction-related genes were identified in different stages. For example, many studies have shown that *BMPR1B* is associated with litter size in animals [32,33,34]. It is the main gene that affects the ovulation rate and litter size in sheep [35,36,37]. In addition, a series of studies have confirmed that the *BMPR1B* can affect follicular development, ovulation [38], and cause ovarian insufficiency [39]. These genes may directly affect the fertility of Boer goats under strong artificial selective pressure at different stages of Boer goat. In addition, *KIT* is verified to be involved in mammalian oocyte growth and follicular development [40]. A lack of *KIT* can lead to the loss of oocytes. In the ovaries of rodents, *KIT* displayed an important role in the migration, proliferation, and survival of primordial germ cells [41].

Among melanin-related genes, *KIT* is pivotal in the melanogenesis signaling pathway, and mutations or deletion of *KIT* can cause different hair and skin colors in mammals [42,43]. For example, a single A-G base missense mutation in exon 13 has been reported to cause differential expression of *KIT* in Liaoning cashmere goats, resulting in different coat colors [44]. This may be the genomic evidence for the changes in coat color in Boer goats at different stages.

Subsequently, excellent growth performance is one of the important goals of Boer goat breeding. Therefore, investigating the selective genes between different breeding stages of Boer goats may help elucidate the genetic basis of muscle development [15]. Notably, some genes identified in this study attracted the attention of the authors. For example, as an essential for maintaining skeletal muscle mass and protein homeostasis, the *RBFOX2* was confirmed to control the fusion of myoblasts in myogenesis by coordinating the alternative splicing of Mef2d and Rock2 [45]. Notch activators (*EDNR*) are putatively involved in the interconnected signaling networks that control satellite cell function [46]. Studies have shown that *EDNR* gene expression is closely related to marbling levels in larger beef cattle populations [47]. In addition, it has also been shown that the delivery of a negative estrogen receptor gene (*EDNR*) abrogates estrogen- and progesterone-regulated gene expression [48].

Finally, the evolution process of animals started from a very distant era, some of the inferior traits of these animals were eliminated, and the superior traits were improved by humans to provide a stable breeding state according to their requirements [49]. Artificial selection resulted in outstanding physiologically and phenotypically changes in these animals [49,50]. This is powerful proof of the coevolution between animal hereditary and feeding environment.

Of course, this study still has a limitation, e.g., the sample size of African Boer goats (10) is observably less than other sample groups, which is also a common flaw in a large number of previous related studies [51,52,53] due to the high sequencing cost. However, high sequencing depth with tenfold genome coverage of each individual and strict quality control conditions were performed in this study can ensure the detection rate of genetic variation enrichment and genotype accuracy of the whole genome.

## 5. Conclusions

This study provided a comprehensive overview of genomic variations in Boer goat genomes under different stages. The characterization of population structure and genomic diversity will point out the direction for genetic assessment and the development for reasonable breeding strategies of Boer goats. Moreover, a series of candidate genes that may be important for the meat quality traits as well growth and fertility of this breed were identified. In addition, some genes also implied that the metabolic and physiological functions of Boer goats are continuously affected by the selective breeding process. These results provide a basis for further resource protection and breeding improvement of this breed.

## Figures and Tables

**Figure 1 animals-12-01356-f001:**
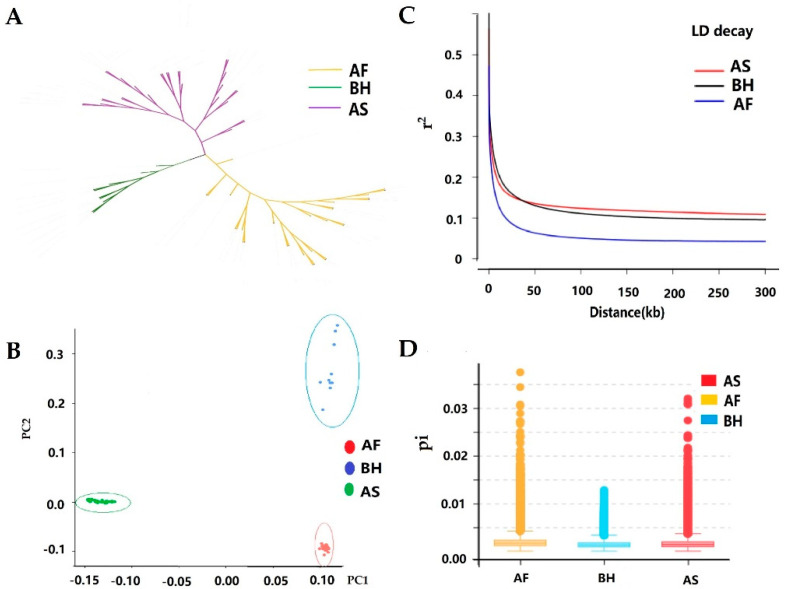
Genomic characteristic of the three goat breeds. (**A**) Neighbor−joining (NJ) tree of the 70 individuals based on the matrix of Hamming genetic distance. (**B**) Plot of the first and the second principal components for the 70 individuals. (**C**) Genome−wide average linkage disequilibrium decay in each breed. (**D**) The boxplot indicates the distribution of nucleotide diversity (pi) of each breed. AF: African indigenous goat, BH: African Boer goat, AS: Australian Boer goat, PC: principal components, NJ: Neighbor−joining, LD: linkage disequilibrium.

**Figure 2 animals-12-01356-f002:**
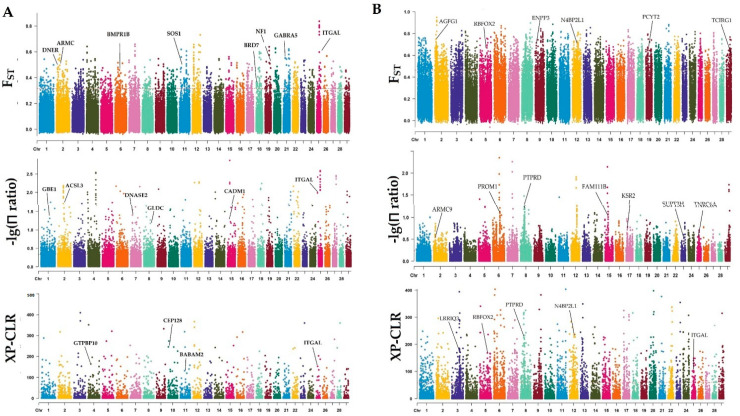
Wide-genome selective sweep analysis for SNPs in 70 goats using a π ratio of nucleotide diversity (π_case_/π_control_), pairwise fixation index (F_ST_), cross-population composite likelihood ratio (XP-CLR). (Manhattan map of F_ST_, Manhattan map of π_case_/π_control_, Manhattan map of XP-CLR.). BH vs. AF (**A**), AS vs. BH (**B**).

**Table 1 animals-12-01356-t001:** The visible divergence of KEGG pathways related digestion and metabolism at two different stages.

	BH vs. AF	AS vs. BH
Digestive system	Gastric acid secretion (*ADCY6*, *ADCY7*)	Protein digestion and absorption (*COL5A3*, *KCNN4*, *COL4A4*)
	Salivary secretion (*LPO*, *ADCY6*, *ADCY7*)	Pancreatic secretion (*SLC4A4*, *PRKACA*, *KCNQ1*)
	Pancreatic secretion (*ADCY6*, *CPA1*, *ADCY7*)	Bile secretion (*SLC4A4*, *ABCG2*)
	Bile secretion (*ADCY6*, *ADCY7*, *LDLRAD4*)	Gastric acid secretion (*PRKACA*, *KCNQ1*)
	Mineral absorption (*SLC30A1*)	Vitamin digestion and absorption (*SLC19A3*)
	Vitamin digestion and absorption (*FOLH1*)	Salivary secretion (*KCNN4*, *PRKACA*)
	Fat digestion and absorption (*NPC1L1*)	
	Cholesterol metabolism (*LDLR*)	
	Protein digestion and absorption (*CPA1*)	
Amino acid metabolism	Lysine degradation (*SETD7*)	Tryptophan metabolism (*KMO*, *ECHS1*)
	Alanine, aspartate and glutamate metabolism (*ASPA*, *FOLH1*)	Valine, leucine and isoleucine degradation (*ECHS1*, *HMGCLL1*)
	Glycine, serine and threonine metabolism (*GLDC*)	Lysine degradation (*ECHS1*, *BBOX1*)
	Cysteine and methionine metabolism (*DNMT1*, *MRI1*)	
	Histidine metabolism (*ASPA*)	
	Valine, leucine and isoleucine degradation (*MCCC1)*	
Metabolism of cofactors and vitamins	Thiamine metabolism (*DDX31*)	Pantothenate and CoA biosynthesis (*ENPP3*)
	Pantothenate and CoA biosynthesis (*DPYS*)	Riboflavin metabolism (*ENPP3*)
	Folate biosynthesis (*GPHN*)	Nicotinate and nicotinamide metabolism (*ENPP3*)
	beta-Alanine metabolism (*DPYS*)	Retinol metabolism (*RPE65*)
Lipid metabolism	Steroid biosynthesis (*MSMO1*)	Fatty acid degradation (*ECHS1*, *ACOX1*)
	Ether lipid metabolism (*PLD1*)	Synthesis and degradation of ketone bodies (*HMGCLL1*)
	Steroid hormone biosynthesis (*DHRS11*)	Glycerolipid metabolism (*MBOAT2*, *AGK*)
	Glycerophospholipid metabolism (*PLD1*)	Fatty acid biosynthesis (*FASN*)
		alpha-Linolenic acid metabolism (*ACOX1*)
		Biosynthesis of unsaturated fatty acids (*ACOX1*)
		Fatty acid elongation (*ECHS1*)
		Sphingolipid metabolism (*SPHK1*)
		Steroid hormone biosynthesis (*HSD17B3*)

AF: African indigenous goat, BH: African Boer goat, AS: Australian Boer goat.

## Data Availability

The sequence data has been deposited at NCBI with the accession number PRJNA671542, PRJNA770516.

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
