# Peer review of "Genome-Wide Selective Analysis of Boer Goat to Investigate the Dynamic Heredity Evolution under Different Stages"

_animals, 2022, doi:10.3390/ani12111356_

Round 1

Reviewer 1 Report

The manuscript can be accepted in the present form.

Author Response

Thank you very much for your review! The revised version of manuscript has been uploaded, please see the attachment.

Reviewer 2 Report

The manuscript still have a very basic weakness that was not solved: only 10 animals of the African Boer breed were included in the study.

This is a too low number to obtain any meaningfull results. Therefore all results obtained with the different methods and comparing 10 vs 30 animals cannot be considered reliable. Therefore the study cannot be considered for publication.

All other interpretations of the results are biased by the starting problem.

English is still very poor, starting from the title and several other parts: wide-genome ? Genome-wide

What is the meaning of parallel genes? this is not a scientific term

It is not clear how the thresholds in all genome wide analyses have been set - this is another critical issue 

Author Response

Thank you very much for your review! The revised version of manuscript has been uploaded,please check!

Reviewer 3 Report

Thank you for addressing the comments and revising the documents.

Author Response

Thank you very much for your review! The revised version of manuscript has been uploaded, please  see the attachment.

This manuscript is a resubmission of an earlier submission. The following is a list of the peer review reports and author responses from that submission.

Round 1

Reviewer 1 Report

The study of Yuan and colleagues describes the identification of the signature of selections and, in general, the genomic characterization of the Boer goat. For this study, the authors used WGS datasets from South Africa indigenous goat (AF, no. 30 animals, from SRA database), African Boer (BH, n. 10 animals sequenced in the present study), and Australian Boer (AS, n. 30 animals from SRA database). Based on the detected SNPs,  the authors carried out pairwise windows-based  Fst and θπ analyses as well as NJ, LD and CVN analyses. Based on the annotation of identified genome regions positive for the selective sweep, the authors described and discussed their main findings. Overall, the manuscript is well written. However, it needs several improvements, especially in the M&M section and in the evaluation of results. For this reason, major revisions are needed.

Introduction.

  • The introduction should report what is known for this breed at the genomic level (e.g. whether genomic studies have been carried out).

M&M should be improved.

  • L71: Are animals in some way related? How did you choose these animals? Add as Supp. Mat. some details about the morphological (or other) characteristics of these animals (coat colour, etc.) as they can be used by the reader to properly evaluate and understand the results.
  • Library size and read length are missing. Please, add a sentence that specifies that the sequencing depth, for BH animals, was expected around 10X. Please, clearly state that identifiers and details of sequencing (for SRA and in house produced data) are presented in Table S1. In Table S1 I do not understand what the field “description” means/reports.
  • L79: what do you mean by “high depth genome datasets” ? A 10X WGS is not considered as high depth.
  • L83: what do you mean with “filtered raw sequencing reads (RSR) were obtained” ? What did you do to your datasets after the retrieval of a quality report? Did you trim reads? Did you discard reads? Please, make it clearer.
  • Please, add the genome accession (GCF_001704415.1). The BWA algorithm used for alignment is missing (MEM, Aln or other?). Please add this info.
  • Which GATK algorithm did you use? (Haplotype caller or other?) Did you call variants considering all the samples together? Please, add this info.
  • Filtering of SNPs is not stated. How did you verify the goodness of the dataset? Did you discard SNPs based on 1) GATK statistics (GATK hard filtering options?), 2) min. DP of sequencing? 3) Is genotype available in all samples or not? Based on the results, did you estimate the Ts/Tv ratio to check that everything is ok? You should compare the Ts/Tv ratio here obtained with the one available from dbSNP. Please, add this info and compute these statistics.
  • Usually, a distance matrix is used to construct a NJ network. How did you carry out NJ here? (the details of Hamming genetic distance is reported in Figure 1. State it in M&M.
  • Any reasons for using a 40 kb sliding windows and 20 kb step size? How many SNPs (average and s.d.) are present in each window? How many windows have been interrogated? Did you discard from the computation those windows presenting few SNPs? How did you choose the best outlier windows (did you use quartiles?, which threshold and how many windows can be selected based on the thresholds?. Please, add all this info and make it clearer. Please, in the results, describe the no. of windows that were retained for FST, phi, and their overlap. I would know if we had a no. of windows in the order of 10, 100 or 1000. It would be better to evaluate the results of each statistic separately and compare them, together, in a second time
  • It is not described from where and how you annotated the windows (GFF?, bedtools? Biomart?)
  • It is not clear how Fst was carried out based on CNV. Please, detail it.
  • Gene enrichment analysis is not well described. Did you use species-specific GO/KEGG terms or did you rely on human or a general one? How many terms have been interrogated for GO and KEGGs? Please, state which multiple testing correction method have been used (Bonferroni, FDR, BH?)
  • I think variant annotation is missing. Did you use VEP or other tools to estimate the fraction of intronic, exonic, missense, stop-gained etc. variants?

Results

  • Describe the statistics of sequencing, please. Are 18321865 a subset of good SNPs? Please, add the Ts/Tv ratio and comment on it. How many of these SNPs are novel compared to dbSNP? Indels have been discarded from these statistics, right?
  • Please, improve the quality of Figure 1 (make it bigger).
  • Stating that 1604 windows are positive for selection signature is a high percentage. It would be better to report the most important. It should be reduced of one order (from 1604 to 160), considering also the top Fst value was around 0.3 (not so high for differentiation). As consequence, stating that more than 500 genes are putatively involved is hard to believe. Probably, it would be better to reduce the threshold from 5% to 1%.
  • It would be better to evaluate and present the results of the two statistics separately, as they have different meanings. And only later, present the results that overlap between the two indexes. Otherwise, specific information is lost.
  • Figure 2 is too small. It is not possible to read gene names and properly evaluate the peaks. Please, make it bigger. Maybe, the rightmost plots (fst vs phi) can be moved to the Mat.
  • L143 Report only data about corrected P-val as having 3033 GO terms is not reasonable. Moreover, this is linked the previous point.

Discussion

Fine.

Reviewer 2 Report

The manuscript reports a simple population genomic study in goats. English and scientific terminology are in general very poor. The experimental design is not appropriate as comparative analyses baseb on only 10 individuals cannot be considered reliable.

Reviewer 3 Report

This paper is nicely written, and organized. Here are some suggestions to improve the paper:

  • Was STRUCTRUE analysis performed across these 3 populations with different K values? It might give us more information about these populations origin and admixture.
  • Sample size: Was the number was enough to compare across 3 groups and do further comparison. What was the power of this analysis?